# Effect of Low- and High-Frequency Auricular Stimulation with Electro-Acupuncture on Cutaneous Microcirculation: A Cross-Over Study in Healthy Subjects

**DOI:** 10.3390/medicines10020017

**Published:** 2023-02-13

**Authors:** Veronica Gagliardi, Giuseppe Gagliardi, Francesco Ceccherelli, Antonello Lovato

**Affiliations:** 1Department of Anesthesiology and Intensive Care, University of Padova, 35122 Padova, Italy; 2Department of Anesthesiology and Intensive Care, Ospedale Santa Maria della Misericordia, 45100 Rovigo, Italy; 3A.I.R.A.S., Via Avellino 11, 35142 Padova, Italy

**Keywords:** vasomotion, electroacupuncture, microcirculation

## Abstract

**Background:** The regulation of microcirculation depends on the dynamic interaction of different factors: the autonomic nervous system plays a pivotal role in the blood flow and acupuncture can modulate it, obtaining different results depending on the site, the frequency, and the intensity of the stimulation. **Methods:** 18 healthy subjects have been enrolled and have undergone two sessions of electroacupuncture stimulations: one session using high frequency and one with low frequency. Microcirculation has been monitored continuously during stimulation using the laser Doppler method. **Results:** The microcirculatory parameters have shown a significant difference between high and low-frequency stimulation, suggesting that low-frequency stimulation is more effective for obtaining a vasodilator effect. **Discussion:** Our results show that low-frequency stimulation can increase the cutaneous microcirculatory flux, without significantly modifying blood pressure and heart rate. The auricular stimulation causes an increase in the activity of the vagus nerve, increasing the cholinergic activity without acting on post-junctional muscarinic receptors. **Conclusion:** Auricular acupuncture has a significant impact on the regulation of microcirculation.

## 1. Introduction

Microcirculation consists of the smallest blood vessels with a diameter lower than 100 microns, including the smallest arterioles, the precapillary sphincters, the capillaries, and the small venules [1]. The rhythmic oscillation in vascular tone caused by local changes in smooth muscle contraction and dilation is called vasomotion [2], a result of the dynamic interaction of sympathetic vasoconstriction, pressure-dependent vasoconstriction, flow-dependent endothelium-mediated vasodilation, metabolic vasodilation, and spontaneous myogenic activity. The vessels implied have spontaneous vasomotion of 2–3 cycles per minute (CPM) and a mean variability in diameter of about 10–20% [2]. In the terminal arteries, the frequency is 10–25 cycles per minute and the width can reach up to 100% of the mean diameter, causing periodic opening and closing of the micro-vessels [3].

The autonomic nervous system (ANS) plays a pivotal role in the blood flow in normal tissues and acupuncture can modulate it. Even though clear evidence about its specific effect is still not well defined [4], experimental evidence demonstrates that acupuncture stimulation can influence the microcirculatory flow by modifying the diameter of the arterioles [5], obtaining different results depending on the site, the frequency, and the intensity of the stimulation. Additionally, the analgesic effects of acupuncture have been well established, and the participation of endogenous opioids and their receptors in Electroacupuncture Analgesia (EAA) has been widely recognized in this framework.

Furthermore, a sympatholytic effect has been demonstrated: the stimulation of lower limbs with low-intensity electroacupuncture (EA) significantly reduces the cardiovascular sympathoexcitatory reflex response from gastric distension in the rat [6,7]. This reflex is integrated at a supraspinal level. The EA on 5–6 Pericardium points (overlying median nerves) activates the neurons of the acuate nucleus, which excites the ventrolateral periaqueductal gray and inhibits cardiovascular sympathoexcitatory neurons in the rostral ventrolateral medulla [8]. Moreover, bradycardia induced by electroacupuncture (EA) is balanced by the administration of atropine, a muscarinic receptor antagonist, owing to the fact that the bradycardia phenomenon is in part caused by the activation of the cardiac innervation deriving from the vagus nerve [9]. Additionally, the stimulation of the Sishencong point (located on the head), increases the cardiac vagal response and reduces sympathetic activity [10].

Auricular stimulation is different from somatic stimulation since it causes an increase in parasympathetic activity without modifying sympathetic activity. This effect develops at the beginning of the stimulation and persists until the end of the stimulation period [11].

In conclusion, the physiological effect of EA stimulation results in a balance between the activity of the parasympathetic and the orthosympathetic systems [10,12].

In this context, the frequency of the stimulation plays a pivotal role: only the low-frequency stimulation is demonstrated to inhibit the sympathetic cardiovascular reflex, suggesting a frequency-specific influence. The underlying mechanisms seem to be related to the quantity of the somatic afferences activated, which is greater when stimulated with the lower frequency stimulation than either the average frequency or the high-frequency stimulation [13].

### Aim of the Study

The aim of this study is to assess the effect of auricular electro-acupuncture at high and low frequencies on cutaneous microcirculation in healthy subjects.

## 2. Materials and Methods

Notably, 18 healthy volunteers were enrolled, 8 of them were males and 10 were females, and their average age was 29.05 ± 5.53 years. They were crossover randomized to be treated with auricular acupuncture both with 2 Hz and with 100 Hz on the auricle. We chose the dominant side for the execution of acupuncture, so we stimulated the right ear of all patients enrolled (we have executed brain lateralization clinical tests on all patients before). This is a randomized crossover study, in which two different sessions of treatment with auricular acupuncture lasting 20 min have been performed, separated by two weeks. The enrolled patients have been randomly assigned to two groups, to start either with the performance of the low-frequency stimulation or with the high-frequency stimulation session. The points stimulated were as follows: Shenmenn point, located on the apex of the triangular dimple, and the thalamus point, located on the inner margin of the antitragus (Figure 1a).

### Experimental Protocol

Each subject has been positioned supine and left in the room without noise for 30 min before the beginning of the auricular stimulation, keeping a constant environmental temperature of 25 °C. Cardiac frequency and pulse oximetry were continuously monitored on the third finger of the right hand. Arterial blood pressure has been measured every 5 min using the non-invasive technique.

Then two probes belonging to a fluxmeter laser-doppler for measuring the microcirculation variations have been placed on the back of both the hands, between the first and the second metacarpal bone (Figure 1b).

After having measured the basal values of vital parameters, the needles for electroacupuncture were positioned (mod Hwuato 25, China) on the aforementioned points, finding the points with a lower electric resistance using the probe (Neuralstift, Svesa 1070, Germany) and linking them through the simulator (mod. Agistim Sedatelec, France). The intensity of the stimulation was gradually incremented depending on the maximal level tolerated by everyone, up to 3 mA during the session. Each session lasted 20 min. The interval between the two stimulation sessions with the different frequencies was 2 weeks.

The cutaneous microcirculatory flux was studied with a fluxmeter laser-doppler (Periflux, PF4000, Perimed AB, Sweden) whose probes were applied on the right and on the left hand between the first and the second metacarpal bone. The Periflux system allows continuous monitoring of cutaneous microcirculation, exploiting the Doppler effect: a laser beam is emitted from the probe to the skin. When the beam hits a moving object, it is reflexed and divided into two components: the first one is reflected with the same frequency as the incident beam, whereas the second component changes its frequency according to the doppler effect, which has been analyzed by the probe. The product of the velocity (v) and red blood cells moving (CMBC) is the flux, expressed as the Perfusion Unit (PU). Another function of the fluxmeter system is the vasomotion analysis.

The activity of each of the factors involved in vasomotion is defined by a specific range of frequency. The analysis of the spectrum of power of vasomotion allows the identification of the role of each regulation factor. The ranges of frequency studied are related to the myogenic activity in the vessel wall (0.052–0.15 Hz), sympathetic activity (0.021–0.052 Hz), and very slow oscillations (0.0095–0.021), which can be modulated by the endothelium-dependent vasodilator acetylcholine [14].

The distribution of the frequency record was studied with “Perisoft for Windows”, reporting the measure of the frequency as cycles per minute. We have studied our data using descriptive analysis to assess the significance level of our results. We calculated the percentage variation between the PU detected at T0 (before the stimulation) and the values observed at T1(after 10 min) and T2 (after 20 min). We performed the statistical analysis using a Chi-squared test, comparing the values observed at the same time in the two different treatments (2 Hz and 100 Hz, respectively).

## 3. Results

During the period of low-frequency stimulation, a variety of cardiac rates up to 10.69% occurred at the end of the stimulation (T2) vs the basal value (T0), associated with non-significant variations of blood pressure (Table 1); the stimulation with 100 Hz neither led to an increase in heart rate nor an increase in blood pressure (Table 2).

Additionally, the microcirculation variation is different when considering the entity of the frequency:

The high-frequency stimulation shows a reduction in PU on both the sites of monitoring −5.95% and −9.65% on the right hand and −8.96% and −10.69% on the left hand at T1 and T2 (Table 3).

The low-frequency stimulation (Table 4) shows an increase in PU. The variation detected is 12.65% and 5.42% on the right hand and 6.05% and 11.74% on the left hand (Figure 2).

### Vasomotion Analysis (Spectral Power Analysis)

To assess vasomotion variations, we compared the oscillation of the PU recorded in the first part of the stimulation (Time 1—10 min) to the PU variation of the second part of the stimulation (Time 11—20 min). During the stimulation, the vasomotion undergoes variations depending on the frequency used.

During low-frequency auricular electrostimulation, the spectral power analysis is significantly different when comparing the first 10 min with the second 10 min: there is an increase in the oscillation of low frequencies in both the right hand and in the left hand (3 cycles per minute: CPM) which are an index of regulation due to neurogenic and endothelial activity [13], whereas myogenic activity has a lower variation (Figure 3a,b).

On the contrary, this variation of vasomotion between the two periods does not occur when considering the high-frequency stimulation (Figure 3c,d).

## 4. Statistical Analysis

We performed a Chi-squared test to analyze the statistical difference between the values of the percentage difference of PU detected at T1 and T2 with low and high-frequency stimulations, respectively. We obtained the results reported in Table 5. Considering the Chi-squared Table, there is a statistically significant difference between the variation of PU in low-frequency stimulation and high-frequency stimulation, with a significance level of *p* < 0.05 (CI 95%) (Appendix A).

## 5. Discussion

Electroacupuncture (EA) represents a specific acupuncture modality that involves electrical stimulation of needles, and its growing use in pain management is supported by scientific research demonstrating differential modulation of endogenous opioids by electrical stimulation of varying frequencies [15] has different systemic and local effects and works along different neurological and hormonal pathways.

Electroacupuncture has an important role in the modulation of autonomic nervous system: it has been demonstrated that Low Frequency EA shows a measurable reduction in sympathetic stress with subsequent improvement in vagal tone. [16]

The frequency of the stimulation is the fundamental parameter, it determines the therapeutic and the regulatory effect of acupuncture.

Many studies have compared the physiological effect of electroacupuncture with different frequency values on different experimental models.

Regarding this, a study compared the effect of electroacupuncture with different frequencies on the latent period and wave amplitude of motor evoked potentials (MEPs) in rats with focal cerebral infarction. It highlighted that the motor function was improved significantly in the 2 Hz electroacupuncture group compared with the other frequency groups (*p* < 0.05), indicating that low frequency electroacupuncture at Shuigou (GV 26) can, with time, improve the latency and amplitude of MEPs after cerebral infarction in rats [17], underlying the important role of low frequency stimulation.

There is evidence that repeated manipulation induced higher local microcirculatory changes that were correlated with the analgesic effects at the relevant sites [18].

There is evidence that repeated manipulation induced higher local microcirculatory changes that were correlated with the analgesic effects at the relevant sites [18]. Our results show that low frequency stimulation can increase the cutaneous microcirculatory flux, without significantly modifying blood pressure and heart rate. These results agree with the published results regarding the somatic acupuncture: the stimulation with electroacupuncture can influence the microcirculation in ovaries [19], of the peripheral nerve [20,21], and other sites, with different effects depending on the site, the frequency, and the intensity of the stimulation.

Wang et al. [22] highlight the key role played by mastocytes in modulating microcirculation after acupuncture: during acupuncture active nerve fibers, traction from the collagen fibers and other factors can activate the mast cells, and through the function of the mast cell band along the vessel, the acupoint will be activated and the microcirculatory response can be observed.

The results of this study confirm the effectiveness of low-frequency auricular stimulation in increasing the flux of peripherical districts, such as somatic acupuncture [18], but the increase of microcirculation flux in this case is a systemic response involving all districts. Moreover, vasomotion analysis highlights that the vagus nerve activity, mostly activated using the auricular low frequency stimulation, is the mechanism responsible for the vasomotion variation due to the endothelial activity. On the contrary, high-frequency stimulation reduces the flux without modifying the vasomotion during the stimulation.

The mechanisms based on the effect of acupuncture on microcirculatory flux are: the axonic reflex, substance P and CGRP (calcitonin gene-related peptide) release from the peripherical nervous terms [20,23], and the pathway of nitric oxide (NO), volatile gas with very low half-life produced by endothelial cells by oxidation of L-arginine by NO-synthetase (e-NOS). e-NOS activity is stimulated by specific mediators (acetylcholine, bradykinin, P substance) or by mechanic stress (shear stress, variations of hydraulic pressure). Once produced by endothelial cells, NO spreads rapidly in the intercellular space to the smooth muscular cells, where induces the release of muscular cells, and consequent vasodilatation. EA stimulation increases the release of NO through the release of cGMP [24], with an increase in plasmatic level 5–60 min after the treatment [18]; vasodilatation and variation of arteriolar diameter disappear after the treatment with *N*(omega)-nitro-l-arginine methyl ester (L-NAME), an inhibitor of e-NOS [25].

The endothelial involvement consequent to the auricular stimulation is fundamental for the effects of EA on microcirculation, according to the vasomotion analysis which detects an increase of frequency due to a higher endothelial activity. This phenomenon is not consequent to the axonic reflex, as described for somatic acupuncture: the results of our study show a systemic response to auricular stimulation with acupuncture, not a localized action. The mechanism of this process has yet to be defined but there may be a neurogenic reflexed cholinergic activity consequent to auricular stimulation with EA. It has been demonstrated that the auricular stimulation causes an increase in the activity of the vagus nerve [26], increasing the cholinergic activity without acting on post-junctional muscarinic receptors [27].

Furthermore, the cholinergic anti-inflammatory pathway (CAP) is activated: it is responsible for the regulation of peripheric immune cells [28].

This pathway of release of acetylcholine could be responsible for the vascular effect of auricular stimulation.

In conclusion, by causing the release of acetylcholine, the action of the vagus nerve on the nerve centers involved and on the immune cells, could well be the main mechanism of the physiological effects detected after electroacupuncture stimulation [29].

## 6. Conclusions

Even though our experiment cannot evaluate the reaction of the whole body, auricular low-frequency acupuncture has a significant impact on the regulation of microcirculation, and better defining the underlying mechanism is important to outline the modalities for its application in clinical practice, using low frequency stimulation. More studies are needed to confirm our hypothesis, but the effect on cholinergic activity of electroacupuncture could well lead to better define the mechanism of action and the rationale for the clinical use.

## Figures and Tables

**Figure 1 medicines-10-00017-f001:**
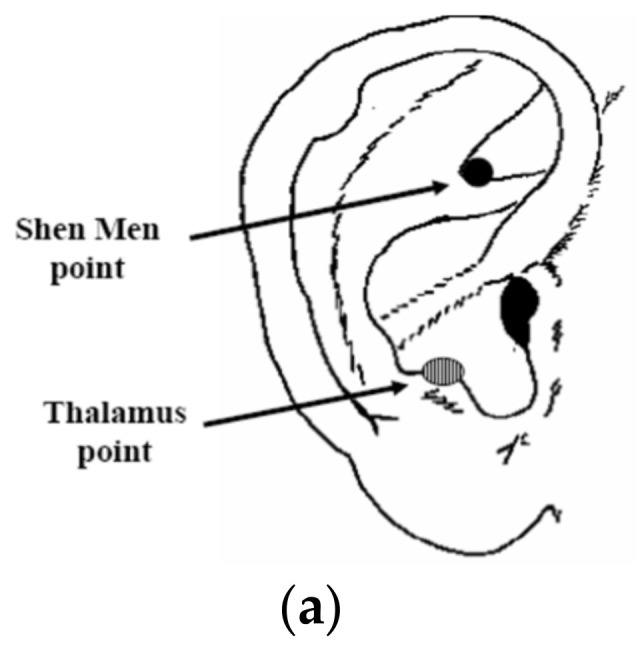
(**a**) Shen Men point, located on the apex of the triangular dimple. Thalamus point, located on the inner margin of the antitragus. (**b**) Fluxmeter laser-doppler (Periflux, PF4000, Perimed AB) probe sites between the first and the second metacarpal bone on both the right and the left hand.

**Figure 2 medicines-10-00017-f002:**
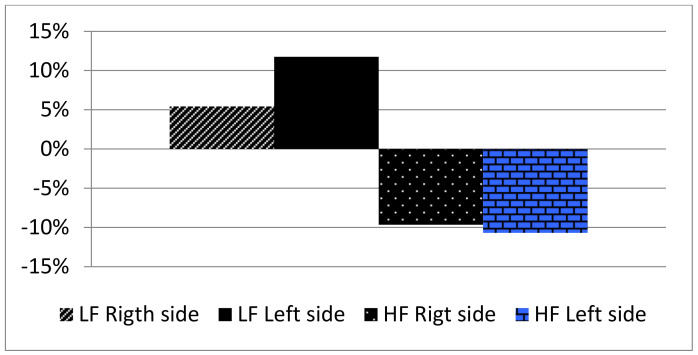
Percentage variation of PU (Perfusion Unit) with high frequency (HF) and low frequency (LF) stimulation.

**Figure 3 medicines-10-00017-f003:**
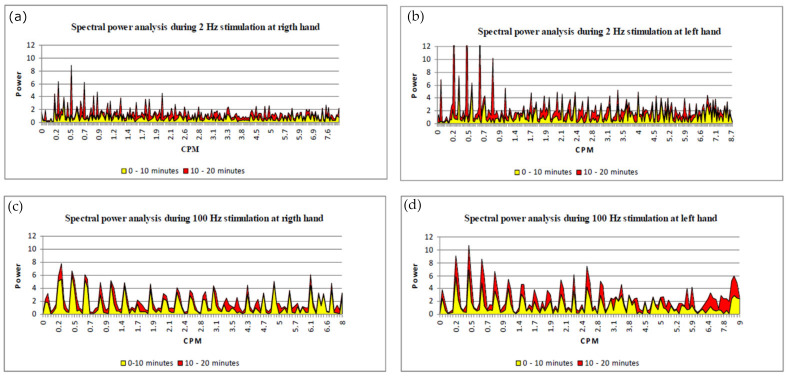
Spectral power analysis (**a**) right hand 2 Hz stimulation; (**b**) left hand 2 Hz stimulation; (**c**) right hand 100 Hz stimulation; (**d**) left hand 100 Hz stimulation. The frequency of the oscillation consequent to the vasomotion variation is measured in cycles per minute (CPM). The yellow areas represent the oscillations occurring during the first part of the stimulation, the red areas represent vasomotion variation of the second part of auricular stimulation.

**Table 1 medicines-10-00017-t001:** Variation of vital parameters (Mean value ± standard deviation) during the low-frequency stimulation (2 Hz): T0: basal time; T1:10 min after stimulation; and T2: end of the stimulation.

2 Hz Ear-Electroacupuncture	T0	T1	T2
Systolic blood pressure (mm Hg)	112.38 ± 1.17	104.23 ± 1.28	108.28 ± 1.18
Diastolic blood pressure (mm Hg)	66.67 ± 0.98	65.28 ± 0.95	66.18 ± 0.90
Heart rate (beats/min)	66.56 ± 1.28	60.44 ± 0.64	59.44 ± 0.64

**Table 2 medicines-10-00017-t002:** Variation of vital parameters (Mean value ± standard deviation) during the high-frequency stimulation (100 Hz): T0: basal time; T1: 10 min after the stimulation; and T2: end of the stimulation.

100 Hz Ear-Electroacupuncture	T0	T1	T2
Systolic blood pressure (mm Hg)	109.34 ± 1.14	111.33 ± 1.22	110.11 ± 1.20
Diastolic blood pressure (mm Hg)	62.64 ± 0.86	64.22 ± 0.89	63.12 ± 0.99
Heart rate (beats/min)	64.42 ± 1.17	65.38 ± 0.59	64.46 ± 0.61

**Table 3 medicines-10-00017-t003:** Mean and SD of microcirculation variation after high frequency (100 Hz) stimulation. T1: basal time; T2: 10 min after stimulation; and T3: end of stimulation. PU is the perfusion unit recorded by the laser-doppler fluxmeter.

100 Hz Stimulation	T0	T1	T2
PU right Mean	10.25	9.64	9.26
SD	4.36	4.87	4.42
% variation vs. T0		−5.95	−9.65
PU left Mean	9.82	8.94	8.77
SD	3.15	3.00	4.07
% variation vs. T0		−8.96	−10.69

**Table 4 medicines-10-00017-t004:** Mean and SD of microcirculation variation after low frequency (2 Hz) stimulation: T0 basal time; T1: 10 min after stimulation; and T2: end of stimulation. PU is the perfusion unit recorded by the laser-doppler fluxmeter.

2 Hz Stimulation	T0	T1	T2
PU right Mean	10.51	11.84	11.08
SD	3.91	5.49	5.79
% variation vs. T0		12.65	5.42
PU left Mean	10.73	11.38	11.99
SD	3.22	4.98	5.98
% variation vs. T0		6.05	11.74

**Table 5 medicines-10-00017-t005:** Results of the Chi-squared test.

T1			T2		
	Expected Values	Observed Values		Expected Values	Observed Values
High-frequency stimulation (200 Hz)	12.65	−5.95	High-frequency stimulation (200 Hz)	5.42	−9.65
Low-frequency stimulation (2 Hz)	12.65	12.65	Low-frequency stimulation (2 Hz)	5.42	5.42
	Chi-squared	1.69886 × 10^−7^		Chi-squared	9.60001 × 10^−11^

## Data Availability

All data generated or analyzed during this study are included in this published article. The data that support the findings of this study are openly available.

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
