# Peer review of "Effect of Low- and High-Frequency Auricular Stimulation with Electro-Acupuncture on Cutaneous Microcirculation: A Cross-Over Study in Healthy Subjects"

_medicines, 2023, doi:10.3390/medicines10020017_

Round 1

Reviewer 1 Report (Previous Reviewer 1)

1. The presented paper describes that low frequency stimulation can increase the cutaneous microcirculatory flux, without significantly modifying blood pressure and heart rate. However, acupuncture is a controversial field of traditional medicine that does not yet have a truly scientific basis. There is no doubt, the research in this field must have significant amounts of proofs and an extremely high level of statistical significance. All of these are lacking in the presented manuscript. 2. The 'Materials and Methods' section is not structured. The important subsection on Statistical analysis is described in only two short sentences. 3. The quality of the figures does not meet the high standards of the journal. 4. It should be noted that the References list does not contain papers in high-impact journals last year. This suggests that the issue has little significance and even may be marginal.

Author Response

I attach the paper with all the amendments suggested about the statistical analysis. In particular, I performed a chi-squared test to analyze the percentage variation. 

Reviewer 2 Report (New Reviewer)

This is interesting and generally clear, although editing of English is necessary for clarity in some places, as set out below. 

How was randomisation done? 

ABSTRACT 

Lines 9-10 - ‘dynamic interaction different factors’ should read ‘dynamic interaction of different factors’ 

Line 10 and line 36 - ‘pivotal role the blood flow’ should read ‘pivotal role in the blood flow’ 

Line 46 - ‘responses’ should read ‘response’ (singluar), and ‘from the gastric distension’ should read ‘from gastric distension’ (ie; no definite article needed here; the whole piece should be reviewed with this in mind). 

Line 48 - ‘activate’ and ‘excite’ should be ‘activates’ and ‘excites’ (it’s the action of the singular EA, not the plural points that is being described)  

Line 49 – similarly, ‘inhibit’ should read ‘inhibits’ 

Line 51 - ‘atropine administration’ should read ‘administration of atropine’ 

Lines 51-2 - ‘consequence of the fact’ should readowing to the fact’ 

Line 53, and again at line 187 - ‘vague’ should read vagus 

Line 54 - ‘reduced’ should read ‘reduces’ 

Lines 56-58 – all the definite articles can be removed here   

Line 58 - ‘also at the end’ should read ‘until the end’ 

Line 61 ‘systems’ should have an abbreviatory final apostrophe, or rephrase as ‘activity of the x&y systems’ 

Page Break 

Line 64 - ‘seems’ should read ‘seem’ 

Line 65 - ‘are a larger number’ should read ‘is greater’ 

Line 70 – remove ‘the’ 

Lines 72, 73, 91 and 94 – in all these instances, ‘have been’ should read ‘were’. Again the whole text wants proofreading with this in mind 

Line 74 - ‘we have chosen’ should read ‘we chose’ 

Lines 75-76 - ‘right ear to’ should read ‘right ear of’. The whole bracketed caveat about the brain lateralisation test should be rewritten. 

Lines 97, 101 and 117 - ‘has been’ should read ‘was’ (again, whole text wants checking for this) 

Lines 106-7 - ‘same frequency of’ should read ‘same frequency as’ 

Line 107 - ‘change’ should read ‘changes’ 

Line 108 - ‘then analyzed’ should read ‘which is then analyzed’ 

Lines 112-3 - ‘allows to identify’ should read ‘allows identification of’ 

Line 119 - ‘and a to assess’ - I’m not sure what you mean here. Just ‘and to assess’? 

Line 120 - ‘we haver performed’ should read ‘we performed’. ‘T student statistical analysis’ should read ‘statistical analysis using a students’ T-test’ 

Line 130 - ‘associated to’ should read ‘associated with’ 

Line 158 - ‘we have compared’ should read ‘we compared’ 

Line 160 – full stop needed after bracket 

Line 180 – again, remove definite articles 

Lines 191-2 - ‘there are’ should read ‘are’ 

Line 192 - ‘the substance P’ should read ‘substance P’ (ie; again, no definite article required) 

Line 196 - ‘idraulicshould read ‘hydraulic’ 

Line 197 - ‘NO spread rapidly’ should read ‘NO spreads rapidly’ 

Line 206 ‘ ‘should not be’ should read ‘is not’ 

Line 220 - ‘rational for the clinical use’ should read ‘rationale for clinical use’ (ie; ‘e’ on the end of ‘rationale’ and no need for definite article) 

Author Response

Dear Reviewer,

I have amended the document following your suggestions. 

Thank you for your worth recommendations

Reviewer 3 Report (New Reviewer)

This study declares whether or not it has accepted an ethics approval, which is a necessary step, otherwise just signing the informed consent is legally risky.

Author Response

I attach the letter of the directory committee, they describe the study and they have acquired the informed consent. 

Round 2

Reviewer 1 Report (Previous Reviewer 1)

1. The presented paper describes that low frequency stimulation can increase the cutaneous microcirculatory flux, without significantly modifying blood pressure and heart rate. However, acupuncture is a controversial field of traditional medicine that has not yet had a significant scientific basis.

2. Figures 1 and 2 should be combined. In addition, they are of poor quality and should be improved.

3. "Figure 3. Percentage variation of PU (Perfusion Unit) with high frequency (HF) and low frequency (LF) stimulation.": The percentage variation has no significance. Need physical values with SD variations.

4. "Table 3.": Mean and SD - ok. Where is N?

5. Table 5: Results of the Chi-square test. The authors should include the p-values for the comparisons for each group. "Chi-squared" has no significance. The statistically significant p-values should be noted in the figures.

Author Response

We are applying techniques which have a demonstrated scientific basis. 

I have tried to improve the images of figure 1 and 2, now they are one figure. 

We have used percentage variation to describe visually the difference between the two types of stimulation. 

I have reported the chi squared table in appendixA 

Reviewer 3 Report (New Reviewer)

This manuscript found a significant impact on the regulation of microcirculation by observing physiological changes in 18 healthy volunteers after alternating high and low frequency electroacupuncture stimulation of ear points.

In this study, healthy volunteers were used as subjects, which is helpful to discover some clinical phenomena and beneficial to the study of physiological mechanisms of acupuncture.

The methodology of this study was correct, the design idea was in line with the actual acupuncture research, the data reporting was complete, and the statistical methods were error-free. However, there is a problem of small sample size, so the scientific significance of this study needs to be further verified.

The discussion in this manuscript is not sufficient, Chinese scholars have done a lot of research on similar topics in the last 40 years and have published a lot of research results on the role of needling at different frequencies in animals and humans. It is hoped that the authors will systematically discuss and analyze the results of this study at both basic and clinical levels.

Author Response

I have updated the reference section. I have added two new articles, 24 and 25, quoting them in the discussion section. 

Round 3

Reviewer 1 Report (Previous Reviewer 1)

The authors have responded to all of my suggestions including adding additional data.

This manuscript is a resubmission of an earlier submission. The following is a list of the peer review reports and author responses from that submission.

Round 1

Reviewer 1 Report

The purpose of the study does not meet the requirements of high scientific quality and significance.